# Climate-Growth Relationships of Chinese Pine (*Pinus tabulaeformis* Carr.) at Mt. Shiren in Climatic Transition Zone, Central China

**DOI:** 10.3390/biology11050753

**Published:** 2022-05-15

**Authors:** Jianfeng Peng, Jinbao Li, Xuan Li, Jiayue Cui, Meng Peng

**Affiliations:** 1College of Geography and Environmental Science, Henan University, Kaifeng 475004, China; lixuan@vip.henu.edu.cn (X.L.); cuijiayue@henu.edu.cn (J.C.); pengmeng@henu.edu.cn (M.P.); 2Henan Key Laboratory of Earth System Observation and Modeling, Henan University, Kaifeng 475004, China; 3National Demonstration Center for Environmental and Planning, Henan University, Kaifeng 475004, China; 4Department of Geography, University of Hong Kong, Pokfulam, Hong Kong 999077, China; jinbao@hku.hk; 5HKU Shenzhen Institute of Research and Innovation, Shenzhen 518057, China

**Keywords:** tree-ring, Chinese Pine, drought variation, climatic transition zone, central China

## Abstract

**Simple Summary:**

Tree rings are widely used in global change research based on the accurate dating capabilities, climate sensitivity and wide distribution of samples. In the context of global warming, the response of tree growths in north–south transition zones to climate change is one of the hot issues in Dendroecology. The research results found that trees’ growth had different responses to May–June temperature and precipitation on the north and south of the mountain. Therefore, we analyzed the relationship between tree ring and a regional hydrothermal composite factor and reconstructed its variation. The variations agree with other drought series and represent the drought variation in central and eastern monsoon regions, and may provide better understanding of drought variation and service for agricultural production.

**Abstract:**

Tree ring data from the southern boundary of Chinese Pine (*Pinus tabulaeformis* Carr.) distribution where is the southern warm temperate margin, the paper analyzes the response of climate factors along north–south direction to tree growth. The results show that temperature and precipitation in May–June and relative moisture from March to June are main limiting factors on trees growth; however, the temperature in the south of the mountains and the moisture in the north of the mountains have relatively greater influence on trees’ growth. Additionally, we also found that the regional scPDSI_MJ_ (that is scPDSI in May–June) was the most significant and stable factor limiting tree growth to be used for reconstruction. The reconstructed scPDSI_MJ_ revealed that there were 29 extremely dry years and 30 extremely wet years during 1801–2016, and it could represent the drought variation in central and eastern monsoon region. The variation exists in good agreement with the reconstructed PDSI for Mt. Shennong and the drought/wetness series in Zhengzhou. Further research found that the droughts of May–June in central China were mainly impacted by local temperature and moisture (including precipitation, soil moisture, potential evaporation and water pressure), and then by the northern Pacific Ocean and the northern Atlantic Ocean. These results may provide better understanding of May–June drought variation and service for agricultural production in central China.

## 1. Introduction

Under global warming, the frequency and intensity of extreme weather events are likely to increase dramatically, which will have a more serious impact on ecosystems and human society and require more timely and effective responses [1]. The typical continental monsoon climate in the middle and lower reaches of the Yellow River can cause the instability and variability of the climate due to the monsoon strength changes and the early or late of monsoon rains. These changes may result in frequent flood and drought in the region and directly related to the agricultural harvests. Hence, the climate changes in the future may not only affect the living environment and ecological security, also put forward severe challenges to the high quality and sustainable development of the middle and lower reaches of the Yellow River region.

There is a long history and abundant climatic records on the middle and lower reaches of the Yellow River, and some reconstructed drought–flood information based on historical documents, but often non-quantitative and discontinuous. Dendroclimatology is a science that reconstructs past climate changes based on tree physiology and tree radial growth characteristics [2,3]. Tree-ring data has the advantages of high resolution (annual or seasonal) and precise-continuous series for dating, and it has been successfully used to quantitatively characterize the climate change series and regarded as one of the most important proxy indicators in the past climate change research [2,4], so it has been used extensively in past global change studies.

In recent years, dendrochronological studies in eastern China have been developing rapidly, including tree ring research of Chinese Pine [5,6,7,8,9,10,11,12,13,14,15,16,17,18,19,20,21]. Chinese Pine is an endemic species and widely distributed in northern China due to its ecological characteristics of cold and drought resistance. The Funiu Mountains is the southern boundary of Chinese Pine distribution, so the growth of Chinese Pine is sensitive to climate change. However, dendrochronological study is still limited in the Funiu Mountains [15,19,22,23,24,25,26] and mainly focuses on hydroclimatic reconstructions, such as temperature [22,24,26,27,28] relative humidity [15,25] and scPDSI [19]. To some extent, the above studies are helpful to understand climate change in the Funiu Mountains and its surrounding areas. To better reveal the impact and formation mechanism on climate change and better understand climate change affecting ecological security and high-quality development in the central plains, further study of tree rings in the Funiu Mountains is necessary.

This study was firstly used to compare correlations between tree-ring data of Chinese Pine and climatic factors (including scPDSI grid data) along north–south gradient in the Funiu Mountains and the transitional characteristics of performance. Secondly, the most significant climate factor limiting on tree growth was selected to carry out climate reconstruction and analyze its spatial representations, and finally further to explore the possibility and potential impact of future climate change in this region.

## 2. Data and Methods

### 2.1. Study Area

Mt. Shiren is located in the Funiu Mountains of the central eastern Qinling Mountains in western Henan province, central China (Figure 1). It is a transitional zone from subtropical to warm temperate continental monsoon climate with mildly wet summers and cold dry winters. It is also the transition zone from humid to sub-humid. The annual mean temperature is 10.0 °C and annual amount of precipitation is 820–860 mm, and mean temperature in January is −3.3 °C, while July is 20.3 °C. The forests canopy cover is about 95%, the dominant vegetation is the mixed forests composed of temperate deciduous broad-leaved and coniferous tree species on the top of the mountain, including Chinese Pine and Huashan Pine (*P. armandi* Franch LC). The soil is typically brown mountain soil with 30 to 40 cm depth. The species for our study, Chinese Pine, is mainly distributed between 1300 m and 1800 m a.s.l. The species is very sensitive to climate change because the eastern Funiu Mountains is the southern and upper boundary of the Chinese Pine distribution [29].

### 2.2. Chronology and Climate Data

Tree-ring data of Chinese Pine is from Mt. Shiren (Mt. Shennong is a contrast site) in western Henan province in July 2017 where had been studied tree growth response to climate factors along west–east gradient (from Luanchuan and Baofeng meteorological stations) in previous studies [15,26]. The sampling site (33°43′39.49″ N, 112°15′4.39″ E, sampling below 1775 m, N-S Ridge) at Mt. Shiren is located in central eastern Funiu Mountains, central China. In general 1 or 2 cores were taken from each tree, and 25 trees/38 cores were sampled using 5.15 mm increment borers. After a lot of works in the lab, the reliable standard chronology from 1801 to 2016 CE, with the starting year determined by adopting a sub-sample signal strength (SSS) threshold of 0.85 [30], was developed and used. The chronology contains 33 cores/21 trees and high signal-to-noise ratio (SNR, 14.543) and the expressed population signal (EPS, 0.936), so it demonstrated a high level of reliability.

To better understand north–south transitional characteristics of tree growth in the Funiu Mountains, the study selected Songxian meteorological station of the northern Funiu Mountains and Neixiang meteorological station of the southern Funiu Mountains (Figure 1, Table 1) along a north–south temperature and precipitation gradient (that is along gradient from warm temperate to subtropical monsoon climate). Climate factors used in this study include monthly mean temperature (T), monthly mean maximum temperature (Tmax), monthly mean minimum temperature (Tmin), monthly total precipitation (P), and monthly mean relative humidity (RH) during 1963–2016 (Figure 2).

The self-calibrating Palmer Drought Severity Index (scPDSI) [31], which represents the severity of dry and wet spells based on monthly temperature and precipitation data as well as the soil–water holding capacity at that location, was chosen over the same time period as a metric to measure the responses of tree growth to moisture conditions. The four grids and a regional mean of the scPDSIs between 33.5 to 34.5 N and 111.5 to 112.5 E (CRU scPDSI 3.26e, https://climexp.knmi.nl/ (accessed on 1 January 2021); Figure 1, Table 1) were also used in this study to compare with climatic data.

### 2.3. Statistical Methods

Pearson’s correlation analyses were performed to identify climate–growth relationships between tree-ring standard chronology and climatic factors from two meteorological stations. Climate–growth relationships were investigated from previous March to current November to explore the potential effects of climatic factors on tree growth from the previous year to the current year.

Then, a simple linear regression model based on the most dominant climatic factor was selected and reconstructed. The split calibration–verification procedure was used to verify the reconstruction [32], statistical parameters for this assessment include correlation coefficient (r), R-squared (R^2^), sign test (ST), reduction of error (RE), and coefficient of efficiency (CE). In general, positive RE and CE indicate a rigorous and reliable reconstruction model [32].

Spectral analyses were performed using Multi-Taper Method (MTM) (Mann and Lee 1996) to detect the periodicities of the reconstruction, and Wavelet analysis [33] was used to extract strong or weak changes for different cycle signals in the reconstruction.

Spatial analyses were performed between the reconstructed scPDSI_MJ_ and scPDSI (1963–2016; 4.05early), temperature, precipitation, potential evaporation, vapor pressure (1963–2016; CRU TS4.04) and soil moisture (1979–2016; CLM/EARi, 0–10 cm) (references for European Climate Assessment & Data) to explore regional representation and possible formation mechanisms. The analyses were performed on the KNMI Climate Explorer (http://climexp.knmi.nl (accessed on 1 January 2021)).

## 3. Results

### 3.1. Climate–Growth Relationship

There are significant negative correlations (*p* < 0.05) with T, Tmax and Tmin in current May and June (Figure 3), and also significant negative correlations with Tmax in current March and April. There are similar significantly negative correlations with temperatures from two meteorological stations, which indicated the strong effect of temperature on tree growth and that the influence time of Tmax to tree growth is longer. In contrast, the chronology shows mostly positive correlations with P in April–May and RH in March–June at both stations. There are also significant positive correlations between chronology and current June P in Songxian station and current July RH from the Neixiang station. Overall, temperature (T, Tmax) of southern meteorological station (Neixiang) of the Funiu Mountains showed higher correlation with tree growth while the greatest influence of humidity (P, RH) were found from northern meteorological station (Songxian) of the Funiu Mountains.

Previous studies demonstrated that the maximum temperature [26] and relative humidity [15] from April to July were main limiting factors at Mt. Shiren. Early summer moisture signal (scPDSI) was also captured and reconstructed in the eastern Qinling Mountains [19].

The correlations between chronology and scPDSI from previous March to current November are almost all positive, especially significant from current March to August. Almost all correlations between chronology and scPDSI in May and June are over 0.6 (*p* < 0.05), the highest correlation in May is scPDSI grid point (0.687, *p* < 0.05; 33.75 N, 112.25 E) closest to the sampling site while the highest correlation in June is regional mean scPDSI (0.663, *p* < 0.05; 33.5–34.5 N, 111.5–112.5 E).

In a general way, the influence of seasonal climatic factor to tree growth is more stable and meaningful than that of single month climate factor. Based on the principle that a higher correlation indicates a greater impact on tree growth, we combine the monthly data on single scPDSI grid point (33.75 N, 112.25 E) closest to the sampling site and regional mean scPDSI (33.5–34.5 N, 111.5–112.5 E). The highest correlations with chronology were both May–June scPDSI, and the highest correlation value of single grid scPDSI was 0.693 (*p* < 0.05, 33.75 N, 112.25 E) while that of regional mean scPDSI was 0.71 (*p* < 0.05).

### 3.2. Transfer Function and Regional scPDSI Reconstruction

According to the above correlation results, the regional mean scPDSI in current May-June (scPDSI_MJ_) was reconstructed using following linear regression equation based on the least square method:scPDSI_MJ_ = 5.514 × Wt − 5.21(1)where, Wt is the index of tree-ring chronology for year t. (N = 54, r = 0.71, R^2^ = 50.4%, R^2^_adj_ = 49.4%, F = 52.795, *p* < 0.0001).

The reconstructed scPDSI_MJ_ could explain 50.4% of the variance in the instrumental record and 49.4% after an adjustment for the loss of the degree of freedom. A visual comparison also showed that the reconstructed mean scPDSI_MJ_ tracked the observed scPDSI_MJ_ well (Figure 4a). The first-order difference data show that there was a significant correlation (r = 0.737, *p* < 0.001) between the two scPDSI_MJ_ sequences (Figure 4b), indicating that the reconstructed scPDSI_MJ_ captured variations of the observed scPDSI_MJ_ at both high and low frequencies.

The split sample procedure was used to assess the reliability of the reconstruction. Table 2 shows that all parameters used for calibration and verification periods are significant (*p* < 0.01) and RE and CE values are positive, and ST test is also significant (*p* < 0.05) and high F values are in all calibration and verification time periods, which means the model is acceptable for scPDSI_MJ_ reconstruction [32] (Cook et al., 1999).

Spectral analysis results (Figure 5) revealed that the scPDSI_MJ_ reconstruction contained 2.30a, 2.86–2.9a, 3.35a, 3.69–3.83a and 6.43a cycles (*p* < 0.5), indicating potential ENSO (2–7a, El Niño-Southern Oscillation) impacts [34,35]. There are also 34.11a, 49.26a cycles (*p* < 0.01), which may be related to PDO (Pacific Decadal Oscillation) or AMO (Atlantic Multi-decadal Oscillation) [36,37].

## 4. Discussion

### 4.1. Tree Growth Influenced by Temperature, Precipitation, or scPDSI

The sampling site located in the eastern Funiu Mountains, the eastern extension of the Qinling Mountains, here is a transitional belt from subtropical to warm-temperate continental monsoon climate. There are higher temperature and more precipitation in the subtropical zone of the southern Funiu Mountains belong to a humid climate region, while they belong to a sub-humid climate region in a warm temperate zone of the northern Funiu Mountains, so the growth of this temperate tree is different in response to climate factors in the north and south Funiu Mountains.

The correlation results found that tree growth at a higher altitude responds similarly to climate factors on the north and south meteorological stations, as described above, and the temperature and precipitation in May and June are mainly limiting factors. Tree growth is the result of the interaction of temperature and precipitation, and temperature often affects tree growth through its effects on water availability. Therefore, temperature-raised water deficiency induces water stress to suppress cell division and expansion [2] and form narrow rings. However, there are also some different responses in the horizontal scale, of which temperature (T, Tmax) of the southern meteorological station (Neixiang) of Funiu Mountains had greater influence on tree growth, while the greatest influence was moisture (P, RH) from the northern meteorological station (Songxian) of the Funiu Mountains.

To better understand the interaction between temperature and humidity, scPDSI was used for further research. We chose four single grid-points and a 1 × 1 grid-point (33.5–34.5 N, 111.5–112.5 E) regional mean scPDSI near the sampling site as examples to conduct correlation analyses, and found that the correlation results were consistent, with significant positively correlations from current March to August. The regional mean grid-point (33.5–34.5 N, 111.5–112.5 E) scPDSI contains both the sampling site and the Songxian meteorological station, and there is a good correspondence between the response of tree growth to scPDSI and the response to the precipitation of the Songxian meteorological station. This also proved that the tree growths in sub-humid areas were mainly restricted by moisture. However, the highest correlation with chronology was regional mean scPDSI (r = 0.71, *p* < 0.05) in current May–June, while it was 0.675 (*p* = 0.000) with relative humidity and 0.583 (*p* = 0.000) with precipitation in current May–June. In other words, with temperatures rising in May–June, deficiency of soil moisture limits tree growth and produces narrow rings due to a lack of precipitation before the rainy season and a high evapotranspiration rate.

The moving correlation results based on a window from January to December and 24 baselength showed that regional mean scPDSI_MJ_ (Figure 6) was the most significantly and stably limiting factor, and also proved that scPDSI of May–June was the main limiting factor for tree growth, hence it could be used for reconstruction.

### 4.2. Drought Variation of the Reconstructed scPDSI_MJ_

According to the length of the reliable chronology, scPDSI_MJ_ variation at Mt. Shiren for the period 1801 to 2016 was carried out (Figure 7). During the past 216 years, the mean of scPDSI_MJ_ was 0.08, and the standard deviation (σ) was 1.607. To investigate historical drought variations, we defined a wet year as above mean +1σ (1.687) and a drought year as below mean –1σ (−1.527) (Figure 7). There are 29 extremely dry years and 30 extremely wet years, which account for 12.96% and 13.89% of the reconstruction, respectively. The five driest years were 1880 (−2.999), 1835 (−2.795), 1955 (−2.723), 1929 (−2.629) and 1907 (−2.552). Similar results were found in previous studies [15,19], and had been recorded in historical documents [38] (Table 3). However, different drought reconstruction methods and classification of drought levels are the main reasons for the differences in these study results (Table 3). Obviously, the most consistent and severe drought events in these studies were the Dingwu Great Drought (1876–1879 year) in Henan and Shanxi (abbreviated Jin), a mega-drought that caused a great famine; and the extreme drought occurred from spring to autumn and the river and pond dried up in 1929 throughout northern China.

### 4.3. The Spatial Representations of Reconstructed scPDSI_MJ_

Spatial correlation analyses of actual or reconstructed regional mean scPDSI_MJ_ with scPDSI (1963–2016; scPDSI, 4.05early), temperature, precipitation, potential evaporation, vapor pressure (1963–2016; CRU TS4.04) and soil moisture (1979–2016; CLM/EARi, 0–10 cm) were performed to determine the spatial distribution and explore possible causes of drought/wet variations.

#### 4.3.1. Spatial Representations

Figure 8 demonstrates that the actual and reconstructed regional scPDSI_MJ_ have significant positive correlations with scPDSI (1963–2016; scPDSI, 4.05early) around the study area. Obviously, the reconstructed scPDSI_MJ_ may represent the drought/wetness variation in central and eastern monsoon region, especially in the central China region and the southern adjacent area. These regions are the main grain producing areas in China, so it is very important to research drought/wet variation for grain production security in China.

In order to verify the reliability of regional representation, we compared reconstructed scPDSI_MJ_ with the reconstructed PDSI of Mt. Shennong (SNPDSI, Figure 1) and drought/wetness index from Zhengzhou (DWZZ [39]; Figure 1 Zhengzhou). The results found that the severe drought events showed better consistency, and correlation coefficients are 0.367 (*n* = 210, *p* = 0.000) between scPDSI_MJ_ and SNPDSI, 0.196 (*n* = 200, *p* = 0.005) between scPDSI_MJ_ and DWZZ, and 0.190 (*n* = 200, *p* = 0.007) between SNPDSI and DWZZ (Figure 9). These statistically validated the reliability and spatial representation of the reconstructed scPDSI_MJ._

#### 4.3.2. Causes analysis of Drought Variations

The drought variation of a region is closely related to regional climatic elements. The spatial correlation results show significant negative correlations with temperature and potential evaporation and significant positive correlations with precipitation and vapor pressure and soil moisture in central China region and south adjacent region (e.g., Huanghuai region and the central-western Jianghuai region) (Figure 10a–j), however the scale is slightly different. In terms of the perspective of affected area, temperature and soil moisture on north and south sides of the sampling site (Mt. Shiren) have similar influence on tree growth (Figure 10a,b,i,j), while precipitation, potential evaporation and water pressure from the north of mountains have more influence than the south of mountains (Figure 10c–h). These also show that drought of central China in May–June is probably mainly impacted by local temperature and moisture (including precipitation, soil moisture, potential evaporation and water pressure) on both sides of the transition zone.

### 4.4. The Global Hydro-Climatic Signals in Reconstructed Mean scPDSI_MJ_

Based on the significant characteristics of monsoon climate in eastern China, this study continues to discuss the relationship between tree growth at sampling sites and global sea surface temperature (SST, NASA MERRA-2 Tsfc, 1980–2016; the relationship between tree growth and land surface temperature has been analyzed in Figure 10a,b). Figure 11 shows there are some significant negative correlations with SST from the northern Pacific Ocean and the northern Atlantic Ocean and significant positive correlation with SST from the southeastern Pacific Ocean and the northeastern Indian Ocean with reconstructed scPDSI_MJ_ in this study, but the overall correlation is weak. These results also confirm that the periodic changes (2.3–6.43a ENSO cycles (*p* < 0.5); 34.11 and 49.26 PDO or AMO cycles (*p* < 0.01)) of tree growth in the transition zone could be related to the SST changes in the Pacific and Atlantic Oceans. The conclusion, which the drought from May to June had a certain relationship with SST from the northwestern Pacific Ocean (that is PDO cycle), was similar to previous studies [40,41], and same as spectral analysis previously. Example, the monsoon from the Pacific Ocean usually arrives at the middle and lower reaches of the Yangtze River in June, and the monsoon rainy season begins in the study area in late July, so the study area in May–June is a period of low rainfall. The high temperature of the northwestern Pacific Ocean surface in May–June reduces the thrust of the monsoon, making it difficult for water vapor to reach the northern China, while heat and evaporation on land lead to water shortages for limiting tree growth (Figure 10a,b) and forming narrow ring. The spatial correlation values from Figure 11 also show that local continental temperature influence (above 0.6) is higher than that of the Pacific Ocean surface temperature (from 0.3 to 0.5), this also indicates that sea surface temperature changes in the Pacific Ocean have a greater impact on tree growth in the study area, but the impact is weaker than that on land in May and June.

## 5. Conclusions

Based on the sampling site located in an ecological sensitive area from north subtropical to warm temperate climatic transition zone, we developed a 216 years ring-width reliable chronology and the correlations between chronology and north–south longitudinal climatic factors were conducted. The results found that temperature and precipitation in May-June and relative humidity from March to June were main limiting factors; however, the temperature in the south of the mountains and the moisture in the north of the mountains had greater influence to tree growth.

The study also showed that regional scPDSI_MJ_ was the most significant and stable limiting factor to be used for reconstruction. The reconstructed scPDSI_MJ_ revealed that there are 12.96% extremely dry years and 13.89% extremely wet years during 1801–2016 year and there existed the five driest years over the reconstructed period. The reconstructed scPDSI_MJ_ can represent the drought variation in central and eastern monsoon region and has a good agreement with other drought sequences in the surroundings.

Further research found that drought of central China in May–June is mainly impacted by local temperature and moisture (including precipitation, soil moisture, and potential evaporation and water pressure) on both sides of the transition zone, and the second one is by the Pacific and the Atlantic. These results may provide better understanding of May–June drought variation and service for agricultural production in central China.

## Figures and Tables

**Figure 1 biology-11-00753-f001:**
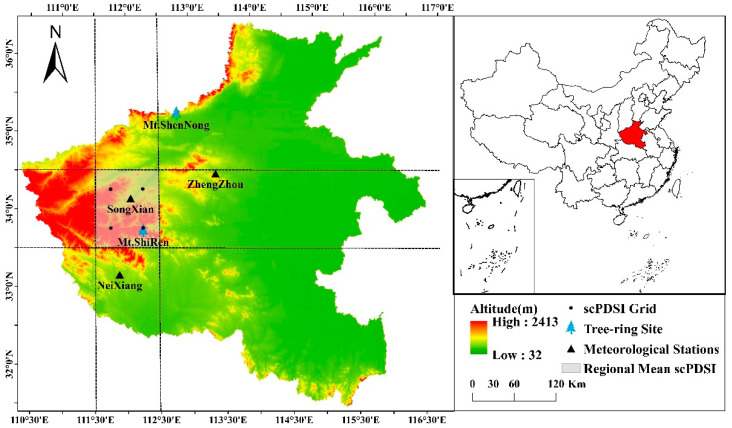
Map showing the study area (Mt. Shiren, 33°43′39.49″N, 112°15′4.39″E), Songxian and Neixiang meteorological stations, the scPDSI grid points and regional mean value, and two contrast sites (Mt. Shennong: tree-ring reconstruction site, Zhengzhou: historical document site).

**Figure 2 biology-11-00753-f002:**
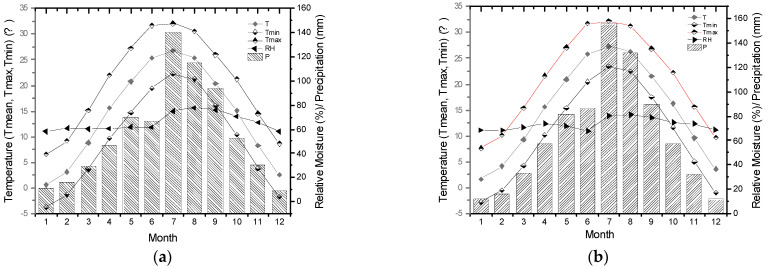
Average annual values of Tmax, T, Tmin, P and RH from Songxian (**a**) and Neixiang (**b**) meteorological stations during 1963–2016.

**Figure 3 biology-11-00753-f003:**
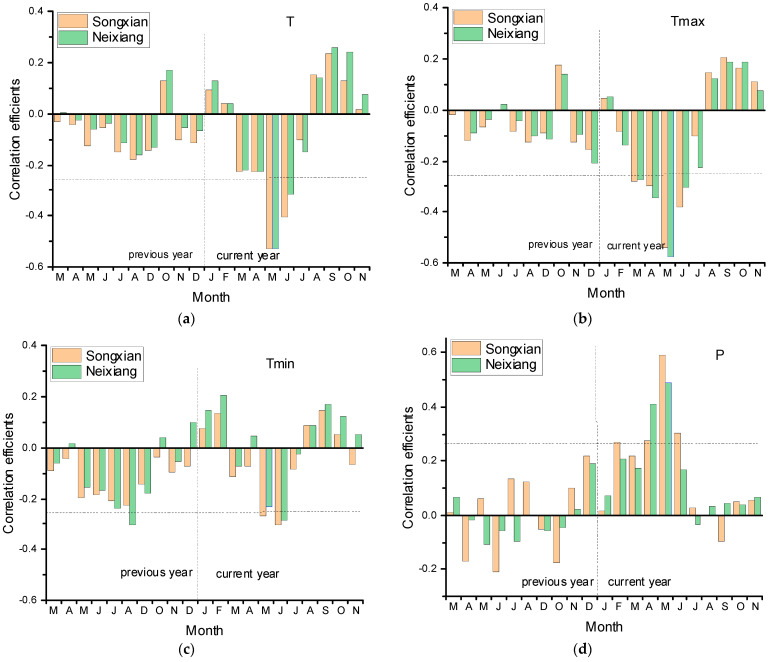
Correlation coefficients between the chronology and climatic factors (T (**a**), Tmax (**b**), Tmin (**c**), P (**d**), RH (**e**) and scPDSI (**f**)) in the Songxian (brown bar) and the Neixiang stations (green bar), respectively. The horizontal dashed line represents *p* < 0.05.

**Figure 4 biology-11-00753-f004:**
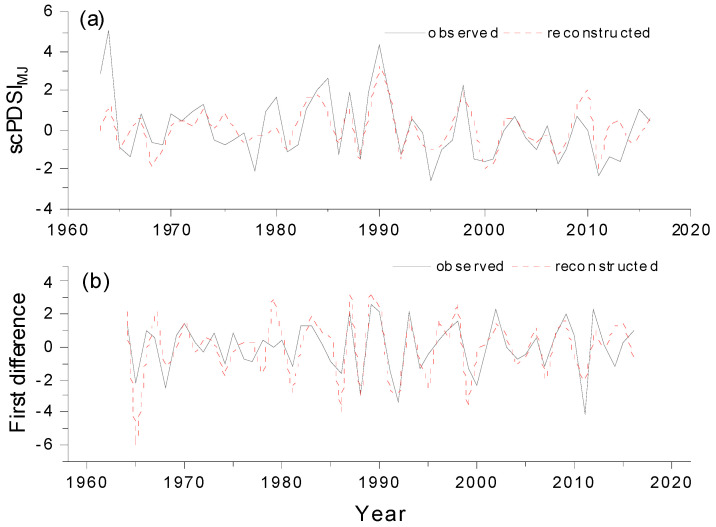
(**a**) Comparison of the observed (solid line) and the reconstructed (dash line) scPDSI_MJ_ during the period 1963—2016 CE at Mt. Shiren. (**b**) Comparison of the first-order difference sequence of the observed (solid line) and reconstructed (dash line) scPDSI_MJ_.

**Figure 5 biology-11-00753-f005:**
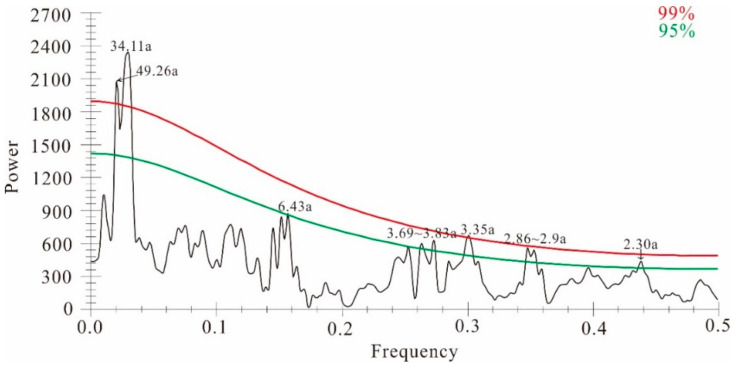
Results of the MTM power spectrum from reconstructed scPDSI_MJ_ from 1801 to 2016 CE. The red and cyan lines indicate 99% and 95%, respectively. “a” in 2.30a on the curve stands for “annual”.

**Figure 6 biology-11-00753-f006:**
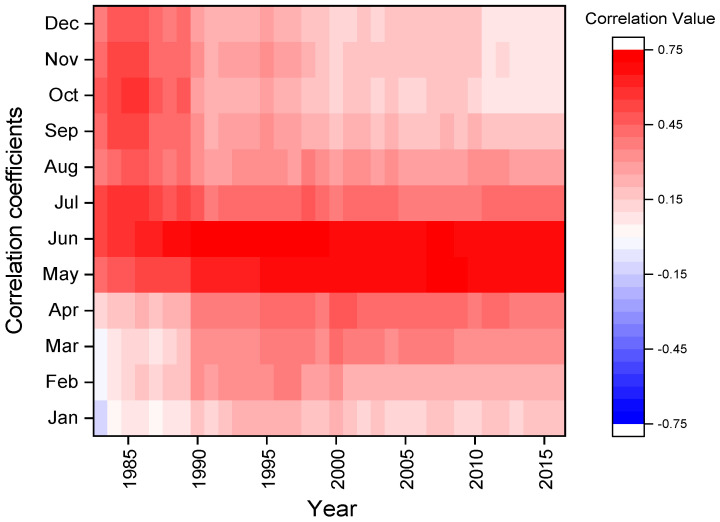
The moving correlation results of regional mean scPDSI_MJ_ based on 24 baselength.

**Figure 7 biology-11-00753-f007:**
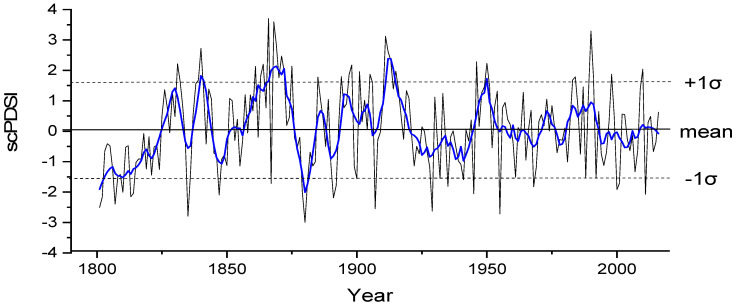
Reconstructed scPDSI_MJ_ and 11 year smoothing curve (blue line) at Mt. Shiren during 1801–2016. The thick and dotted horizontal lines represent mean scPDSI_MJ_ and its standard deviation, respectively.

**Figure 8 biology-11-00753-f008:**
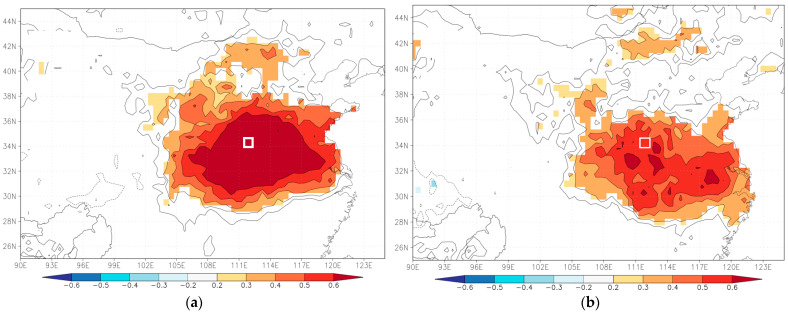
Spatial (90°–126°E, 25°–45°N) correlation between actual (**a**) and reconstructed (**b**) scPDSI_MJ_ and scPDSI (1963–2016; 4.05 early). The white rectangle represents the reconstructed scPDSI_MJ_ area (33.5–34.5 N, 111.5–112.5 E).

**Figure 9 biology-11-00753-f009:**
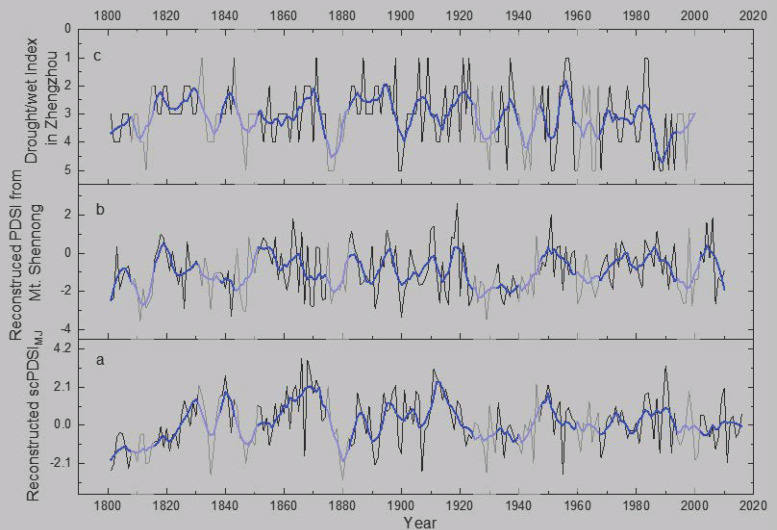
Comparisons of (**a**) reconstructed regional scPDSI_MJ_, (**b**) reconstructed PDSI from Mt. Shennong [12], and (**c**) drought/wet index in Zhengzhou [39]. The gray bars represent the same drought periods in three series.

**Figure 10 biology-11-00753-f010:**
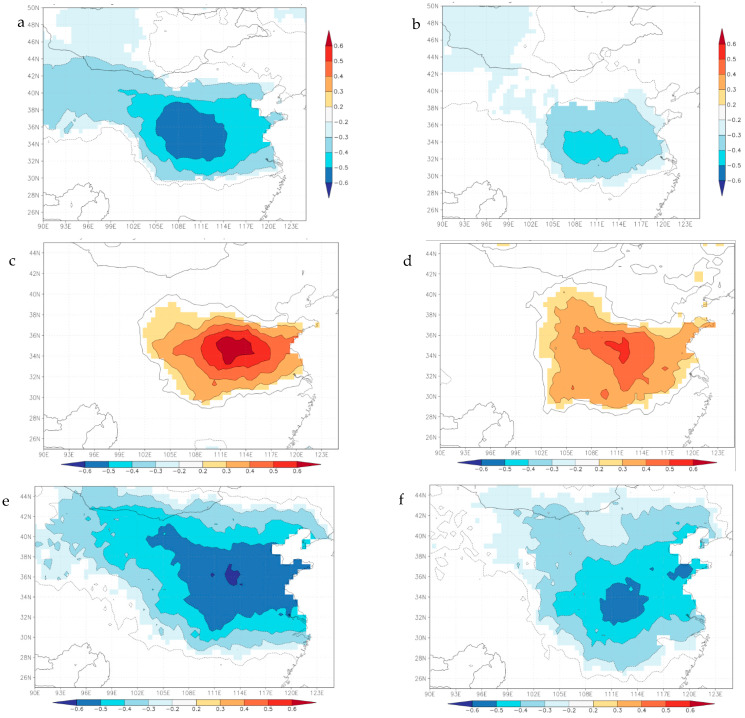
Spatial (90°–126°E, 25°–45°N) correlation analyses between actual (left) and reconstructed scPDSI_MJ_ (right). (**a**,**b**) temperature, (**c**,**d**) precipitation, (**e**,**f**) potential evaporation, (**g**,**h**) vapor pressure (CRU TS4.04, 1963–2016), and (**i**,**j**) soil moisture (1979–2016, CLM/EARi, 0–10 cm).

**Figure 11 biology-11-00753-f011:**
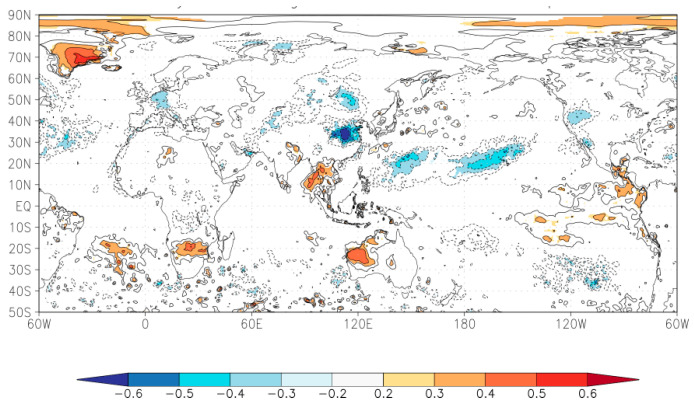
The relationships between tree growth and global sea surface temperature (SST (60° W–120° E–60° W, 50° S–90° N), NASA MERRA-2 Tsfc, 1980–2016).

**Table 1 biology-11-00753-t001:** Characteristics of climate data used in correlation analyses.

Climate Data Source	Latitude (°N)	Longitude (°E)	Elevation (m a.s.l.)
Songxian	34.13	112.06	440.7
Neixiang	33.15	111.88	221.4
scPDSI-1	33.75	111.75	-
scPDSI-2	33.75	112.25	-
scPDSI-3	34.25	111.75	-
scPDSI-4	34.25	112.25	-
Regional scPDSI	33.5–34.5	111.5–112.5	-

**Table 2 biology-11-00753-t002:** Calibration and verification statistics for reconstructed scPDSI_MJ_.

	Calibration (1963–1991)	Verification (1992–2016)	Calibration (1988–2016)	Verification (1963–1987)	Full Calibration (1963–2016)
R	0.714 **	0.696 **	0.814 **	0.601 **	0.710 **
R^2^	0.510	0.485	0.662	0.361	0.504
CE		0.298		0.324	
RE		0.668		0.440	
ST	22+/7− **	18+/7− *	22+/7− **	19+/6− *	41+/13− **
F	28.108	21.663	52.948	12.994	52.795

** Significant at the 99% confidence levels, * Significant at the 95% confidence levels.

**Table 3 biology-11-00753-t003:** Moderately to extreme drought/wet events derived from tree-ring reconstructions and the corresponding descriptions of historical records.

	scPDSI_MJ_ (1801–2016, This Study)	RH_AJ_ (1801–2016) [15]	scPDSI_MJJ_ (1868–2005) [19]	Historical Records in Henan Province [38]
Extremely dry period	1801–1802	1801–1802		Not available
	1807,1810	1807–1810		Drought from spring to summer
	1813–1814	1813–1814		Drought from spring to summer in 1813 and spring drought in 1814
		1821		Not available
	1835	1835–1836		Severe drought from spring to summer
	1847	1847		Drought from spring to summer
	1867	1867		Drought from spring to summer in Gongyi, Henan
	1879–1881	1879–1881	1879	A mega-drought caused a great famine over Henan and Shaanxi so on provinces in northern China in during 1876–1879
	1891–1892	1891–1892		Not available
	1900	1900	1900	Severe drought from spring to autumn over Henan and Shanxi
	1907	1907		Drought from spring to summer
			1923,1926,	Not available
	1929	1929	1929	Severe drought from spring to autumn over Henan, the river and pond dry up
	1932	1932		Drought from summer to autumn over Henan
	1935	1935		Severe spring drought over Henan
	1941	1941		Severe drought from spring to autumn in 1941 and 1942
	1945	1945		Severe drought and locust disaster over Henan
	1955	1955		Severe drought from spring to early summer
	1968	1968,		Drought from spring to summer in the most of Henan
	1988	1988		Severe drought following the drought of 1985–1987
	1992,	1992,		Severe spring drought
			1994,1995	Not available
	2000–2001	2000–2001	2000	Severe drought from February to May (the worst one since 1950)
		2007		Not available
	2011	2011		Severe drought from January to February since October 2010 (the lowest for the same period since 1951)
Extremely wet period	1831			Spring and summer rains from north to south of the Yellow River
	1840			Not available
	1862–1864			Not available
	1866			Flood in March in Yexian
	1868–1872		1869	Flood in summer and autumn over Henan in 1869
	1875			Not available
			1883	Not available
	1885		1885	Flood in summer in Lingbao and Shanxian (northwestern Henan)
	1894		1894	Not available
			1895	Not available
	1898		1898	Severe flood in summer at Yi and Luo river (Henan), Shangnan (Shaanxi)
	1901			Flood in Fangcheng (Henan)
	1905		1905	Severe flood in spring and summer over Henan
			1906	Not available
	1911–1913		1910–1912	Persistent flood in summer and autumn over Henan during 1910- 1913
	1915			Flood in summer and autumn over Henan
			1933,1934,	Not available
			1936,1944	Not available
	1946			Persistent rainfall in spring and summer in the most of Henan
			1948,1949	Not available
	1950			Not available
			1973,1980,	Not available
	1983		1983	Rainstorm in April and May in the north Henan in 1983
	1984		1984	From June to September, there were 5 large-scale rainstorms over Henan in 1984
	1990		1990	Not available
	1991			Rainstorm in September in Nanzhao (Henan)
	1998,			Not available
	2010			Low temperature and rain in spring, heavy rain in summer
The five driest years	1880 (−2.999)	1880 (57.82)	1879 (−3.61)	Ding-Wu disaster, Shanxi and Henan famine, extreme drought
	1835 (−2.795)	1835 (60.05)	2000 (−2.94)	
	1955 (−2.723)	1955 (60.24)	1929 (−2.53)	
	1929 (−2.629)	1929 (60.50)	1926 (−2.33)	
	1907 (−2.552)	1907 (60.71)	1923 (−2.28)	

## Data Availability

Not applicable.

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
