# Peer review of "Climate-Growth Relationships of Chinese Pine (Pinus tabulaeformis Carr.) at Mt. Shiren in Climatic Transition Zone, Central China"

_biology, 2022, doi:10.3390/biology11050753_

Round 1

Reviewer 1 Report

This research is fine, though revision of the manuscript is needed to improve it before publication.  Acknowledging how difficult it is to write in a non-native language, the English grammar and style needs editing.  Right off the bat, the title is awkward stylistically.  It’s not a reviewer’s place to point out grammar and style errors, but a pro English stylist would be of great use here.

Otherwise, a few comments on the content:

Section 2.1: This seems to refer to Figure 2, but there’s no (Fig 2) reference here.

Section 2.2: Citing a previous publication for Methods details is done in scientific articles, but this ms has little to no detail on the many steps that go into the construction and analysis of tree-ring chronologies.  Sample depth?  COFECHA signal strength?  SNR, by the way, is problematic as an indicator of signal strength because it is dependent on sample depth, i.e., the more trees sampled, the higher the SNR, even if the actual signal isn’t very strong.  Rbar is not affected this way and would be better.

Table 1: A whole column of the same data (1963-2016) would indicate that that column isn’t necessary.

Figure 3: It is typical to try (and present) correlations of multi-month seasons, in addition to single months.  Meko’s SEASCORR app would be perfect for this approach.  It might be that even months that are not significant by themselves would be included in longer multi-month seasons.

Figure 3: It looks as if pre-season P would be the strongest individual contributor to a dendroclimatic model here, as it often is.  But the emphasis in the text seems to be more on temperature.  Yes, T is inversely related to water availability, but water availability to trees really starts with rainfall.

Line 158: So, the season of choice here is May-June?  In Peng et al. (2020), the precursor paper to this one, it was April-July.

Figure 4: It would be helpful to add a horizontal reference line to time series plots, 0.0 in both of these cases.

Figure 7. The very first sentence of the Introduction invoked global warming as a motivation for this research.  In looking at this reconstruction of PDSI, is there any unexpected departure during recent decades that might be considered evidence of global warming (or any change)?

Reviewer 2 Report

Comments and Suggestions for Authors

This is the review of the manuscript

Journal: MDPI Biology

Manuscript ID: biology-1679906

Authors: Jianfeng Peng, Jinbao Li, Xuan Li, Jiayue Cui, Meng Peng

Title: Climate-growth relationships from tree-ring in the southern boundary of Pinus tabulaeformis from Mt. Shiren, central China

The authors study climate-growth relationships of Pinus tabulaeformis on Mt. Shiren in central China and reconstruct PDSI for the May-June period for 1801-2016.

The article is very interesting due to the very current problem of the currently observed climate change.

I have few comments and suggestions to authors.

Below I list specific comments:

Basic note: why such a large part of the results is included in the section: discussion. In this section you should link / compare your results with the results of other authors, but there you present the results of your analyzes and refer to only a few publications.

line 33 - IPCC reports are newer and can be used

line 51 here is Cheng et al 2012, 2015, 2021 and there is Chen in the bibliography ....

line 79 - for the first time the whole word Pinus, and then the abbreviation P.

Location of the research area - you say once about MT. Shiren, and once for Mt. Funiu, you often use these names interchangeably, I do not know this area, so write it so that a person from outside China knows where your research area is.

Figure 1 - why on the upper right map you show zoom of southern China ??, move the scale to a colored map, Why the map shows the location of Zheng Zhou and Mt. Shen Nong ??, on the map and in the text other names SongXian (Songxian, line 97) and NeiXiang (Neixiang), unify it

line 88-96 - once again Mt. Shiren and then Mt. Funiu, write more clearly,

there is a lack of a lot of data on the chronology: how many trees it was used to built of chronology, at what height the trees grew, what location of the slope to the sides of the world and to the direction of the incoming air masses, indexing method…

Figure 2 - use the colors in this drawing, the rainfall shoud be brighter, because the RH waveform is not visible, in the caption explain what the abbreviations and other names of the stations in Fig. 1 and 2

line 123 - other authors in the bibliography are here: Mann and Lee, but there ... and Lees

line 143-144: of southern meteorological station of Mt. Shiren, but in line 98: and Neixiang meteorological station of southern Mt. Funiu, I don't know where your research area is anymore !!! in line 143 give the name of the station

line 145 - give the name of the station

figure 3 - why some bars have a blue border ??, on part (d) - you can not see the colors, you can only see them in the legend, mark the previous year on the axes, e.g. pM, pA

line 168 - nowhere has it been stated how the indexed chronology and here you  use the indexes

Table 2 - transfer on 1 page and explain all abbreviations used in the caption

figure 5 – explain what mean MTM

line 191 - what does 2-7a mean ??

line 192 - Sun and Wang 2007 - differently in the bibliography

line 198-205 - this is part of the discussion, but lines 206-237 are your results (again confusion: Mt. Funiu and Mt. Shiren)

lines 244-255 - style unnecessarily as in the figure caption, only from line 249 - this is a discussion, earlier these are the results

Table 3 - why are some dates in bold? CCMDC 2006 is different in the bibliography, or for 1990 and 1998 for sure no data ?? data not available ??

lines 279-301 this are the results

Figure 9 - Peng at al 2012 - no in bibliography

line 306-354 this are the results

Figure 11 what does tree growth mean ??

References

Chen et al 2012, 2015, 2021 not quoted in the text

Daniels 2020 - not quoted in the text

Su and Wang - otherwise in the text

Reviewer 3 Report

The manuscript of Jianfeng Peng, Jinbao Li, Xuan Li, Jiayue Cui, Meng Peng “Climate-growth relationships from tree-ring in the southern boundary of Pinus tabulaeformis from Mt. Shiren, central China” describes dendroclimatic analysis and reconstruction of PDSI drought index in Henan province of central China, based on standard tree-ring width chronology of Chinese pine. This is typical case study; its topic is of interest mostly at regional/country scale. Used scientific methodology is valid; however, there are some concerns. First, used series of PDSI are available from 1901, but by cutting those off from 1963 authors could reduce reliability of their reconstruction. It is valid approach to show all dendroclimatic correlations for the same cover period for comparison which climatic variable impacts tree growth more. But for the sake of reconstruction, maximal length of actual PDSI series should be considered.

Quality of English translation is insufficient; many sentences are unclear or have stylistics and grammar errors. Some examples of errors or sentences difficult to understand are presented below, in minor comments. It is strongly recommended to improve quality with help of appropriate services or fluent English speaking colleague.

Title and Abstract reflect content of the manuscript.

Introduction.

Relevance of study and state of art are presented adequately.

Materials and Methods.

In description of the study area, mean annual and monthly values of temperature and precipitation are stated. You should explicitly tell data source (climatic station(s), gridded data CRU TS for particular coordinates, etc.) and years of averaging for these values. In description of chronology, method of standardization is not stated, and most of chronology statistics are omitted. It is recommended also to show this chronology and sample depth as figure. For usage of PDSI, the reason of four grid point usage is not stated, but I can assume from results that it was used to try and find spatial scale that results in the best correlation with considered chronology and the best fitness of reconstruction model.

For spectral analysis, I would recommend performing it on actual PDSI series too (from 1901, mind) and compare cycles found in actual and reconstructed series where their ranges overlap.

For moving correlations of tree-ring chronology with PDSI, I would like to see results for separate grid points too, as proof of your statement that correlation with ‘regional mean scPDSImj (Fig 6) is the most significant and stable’.

Results.

Presented adequately, figures and tables are clear.

Discussion.

This section contains more continuation of results, and less arguments and reasoning. Please rewrite it. Put actual data (description, plots and tables) of quantitative comparison between your reconstruction and other series in results, and only discuss it here. Also, reasoning of climatic response for two stations seems confusing. Trees react to actual local temperature, soil moisture, air humidity, so the question is not what station’s series impact tree growth more! It is rather at which station temperature (precipitation, etc.) is more similar to the climate of sampling site, and/or which factor impacts tree growth stronger.

Conclusions.

Middle part of conclusion repeats results rather than actually concludes findings. Also there is a lot of repetition from abstract.

References.

Please format references and citations in accordance with journal requirements. Links: https://www.mdpi.com/journal/biology/instructions, https://www.mdpi.com/authors/references.

Minor comments

L3, 11, 79, etc. When species mentioned first time in the manuscript title, abstract, and text, there should be full Latin name with species author: ‘Pinus tabulaeformis Carr.’; in all further places it can be ‘Pinus tabulaeformis’ or ‘P. tabulaeformis’. If non-Latin name (Chinese pine) is used throughout the manuscript too (as in L82), it would be wise to put it first time together with Latin name

L10. Delete ‘Corresponding author:’

L15, ____. I recommend to use throughout the text ‘moisture availability’ or ‘moisture regime’ instead of ‘humidity’ (except relative humidity as a climatic variable)

L17. Please rephrase as ‘PDSI is … factor limiting tree growth’

L28. Keyword ‘transition zone’ is not appropriate. It is transition between what and what? Meaning of this term may vary greatly with context

L34, 42. Not sure if the Yellow River name should be translated or stated as ‘Huang’ in Chinese

L36. “These changes result in...’ – delete ‘be’ here

L37. ‘harvest of agricultural years’ – what do you mean?

L46 rephrase as ‘precise-continuous dating of series’

L53 etc. Use ‘Mt.’ for single mountain peak (Mt. Shiren, I presume), and full word ‘Mountains’ for mountain ridges, ranges and larger systems, like ‘Funiu Mountains’. Now usage of abbreviation can confuse readers

L59-62. Sentence is difficult to understand

L64. What do you mean here: ‘north-south climatic factors’? The same in L89 with ‘east-west climate factors’

L65-66. ‘climate-limiting factor’ – did you mean limiting tree growth climatic factor?

L78. ‘forests composed of temperate deciduous broad-leaved and coniferous’ – add word ‘trees’, or ‘species’, or both (‘tree species’)

L80. Replace ‘deep’ with ‘depth’

L93. You cited Wigley et al. (1984), which means that you should use threshold EPS>0.85, not SSS>0.85.

L106. Replace ‘chose’ with ‘chosen’

L109. Delete word ‘respectively’ it is not necessary here

L115. ‘tree growth’, not ‘trees growth’

L143-145. Phrases ‘southern meteorological station of Mt. Shiren’ and ‘northern meteorological station of Mt. Shiren’ are difficult to understand. It is written as if both stations are actually located on this mountain, just on the opposite slopes of it

L154. It should be ‘seasonal climatic response’, not ‘seasonal climate’ that is more stable than monthly one

Figure 4. Switch panels (a) and (b)

Figure 5, L191 and further. ‘2.3a’ etc. – is it length of cycles in years? Not all readers are familiar with such abbreviation

L199. ‘relative humidity’ – write without capital letters

L207-208. Rephrase as ‘a transitional belt from subtropical to warm-temperate continental monsoon climate’

Figure 6. Please state window and step of moving correlations

L248. Use ‘extremely dry’ or ‘driest’, not mix of both

L249-250. Citation does not match: text cites Liu et al. 2017 and Peng et al. 2020, whereas Table 3 cites Zhao, 2019 and Peng 2020

Table 3. I would offer add comparison with actual PDSI series too after 1901 (just add one more column where state years when actual regional seasonal PDSI has values above mean+1σ and below mean-1σ

L286. Replace ‘are’ with ‘have’

Figure 8. The white rectangle mentioned in caption is not presented on figure itself

Reviewer 4 Report

The manuscript “Climate-growth relationships from tree-ring in the southern boundary of Pinus tabulaeformis from Mt. Shiren, central China” presents a study on the relationship between hydroclimatic parameters and growth of Pinus tabulaeformis in a region of China, while a reconstruction of seasonal drought conditions (May-June) based on scPDSI is attempted and discussed. The study is based on a tree-ring chronology from 1801 to 2016, climatic data from two meteorological stations and gridded data (4 points) of scPDSI index.

The topic of the manuscript is timely and interesting, as it may provide information on the identification of the climatic conditions in the study region during the past (reconstructed) period, as well as to investigate the sensitivity of the specific species to climatic variations. The paper is well-written and properly structured, while the methodological approach seems appropriate for the purposes of the study. However, there are points that should be further elaborated, for enhancing the analysis and the validity of the presented work.

More specifically:

  1. [Section 1] The introductory section presents a proper general overview of on the topic of the study. However, there should be also a brief literature review the tree species under study, the main aspects regarding the selected approach justifying its suitability for the specific case, as well as similar cases for pinus species related to the response to hydroclimatic parameters (e.g. see https://doi.org/10.3390/f10090752, https://doi.org/10.3390/atmos11060554, https://doi.org/10.3390/f9070440). It is also noted that the reference to “climate change” concept in some instances in the text (e.g. “… characterize the climate change series …”, “...the studies are helpful to understand climate change…”) is not always accurate, considering that dendrochronological studies can identify past climatic characteristics of region, which does not necessarily indicate climate “changes”.
  2. [Section 2] Please elaborate on tree-ring sampling and handling approach (sampling time, sampling locations’ altitude, number of samples, timeseries, etc.) and indicate the approximate distance between each meteorological station and the sampling sites. The above are rather important for explaining / discussing the identified relationships between tree growth and climatic factors (e.g. Tmax, P).
  3. [Fig. 2] It seems that the figure presents the average monthly values for the study period; please clarify in the figure caption.
  4. [Section 3] Based on scPDSI seasonal analysis, it would be also meaningful to perform a similar seasonal analysis for the other climatic parameters (i.e., cumulative seasonal P, average seasonal T, etc.) for the corresponding period.
  5. [Section 4] There should be some discussion on the possible uncertainties of the analysis, related to sampling issues (e.g. number of samples?), the actual association of the available stations / grid points to the sampling locations (distance, direction, topography, altitude differences, etc.); a commend on the uncertainties would be also appropriate in the conclusion. It must be also clarified and taken into account in the discussion and comparison to other studies, that the reconstructed drought conditions based on scPDSI refers specifically to May-June period, which does not necessarily represent the annual conditions or other seasons of each year. Furthermore, it is not clear how was the spatial analysis of the reconstructed scPDSI performed; please elaborate (this should be in the methodological section).
  6. The language is generally understandable; however an edit of the text for addressing language issues is suggested.
  7. Please adjust the citations in the text according to the style of the journal (reference numbers).

Round 2

Reviewer 3 Report

Authors somewhat improved their manuscript, but many of my comments from the first stage of revision (I) were not worked on satisfactorily, especially comments on grammar corrections, and restructuring discussion conclusion sections. Below are responses of authors to those comments with my second-stage (II) comments.

Author's Reply to the Review Report (Reviewer 3):

Thank Reviewer very much for your comments and suggestions.

Firstly, we have changed title “Climate-growth relationships from tree-ring in the southern boundary of Pinus tabulaeformis from Mt. Shiren, central China” into “Climate-growth response of Pinus tabulaeformis growing climatic transition zone at Mt. Shiren, central China”

Reviewer’s Comment II: ‘growing in climatic transition zone’, otherwise correct.

Reviewer’s Comment I: It is valid approach to show all dendroclimatic correlations for the same cover period for comparison which climatic variable impacts tree growth more. But for the sake of reconstruction, maximal length of actual PDSI series should be considered.

Author's Reply:That's not a very accurate statement. Because the PDSI index is a reconstructed data by other proxy data, there is uncertainty. This study is the result of comparison with climatic data selected over the same time period, so the selection is reliable.

Reviewer’s Comment II: Please mention explicitly this reasoning in the text too, not only here.

Reviewer’s Comment I: Quality of English translation is insufficient; many sentences are unclear or have stylistics and grammar errors. Some examples of errors or sentences difficult to understand are presented below, in minor comments. It is strongly recommended to improve quality with help of appropriate services or fluent English speaking colleague.

Author's Reply:Thanks very much! We have revised.

Reviewer’s Comment II: In many places comments about grammar errors were ignored.

Reviewer’s Comment I: Discussion. This section contains more continuation of results, and less arguments and reasoning. Please rewrite it. Put actual data (description, plots and tables) of quantitative comparison between your reconstruction and other series in results, and only discuss it here. Also, reasoning of climatic response for two stations seems confusing. Trees react to actual local temperature, soil moisture, air humidity, so the question is not what station’s series impact tree growth more! It is rather at which station temperature (precipitation, etc.) is more similar to the climate of sampling site, and/or which factor impacts tree growth stronger.

NOT answered.

Reviewer’s Comment I: Middle part of conclusion repeats results rather than actually concludes findings. Also there is a lot of repetition from abstract.

NOT answered.

Reviewer’s Comment I: L3, 11, 79, etc. When species mentioned first time in the manuscript title, abstract, and text, there should be full Latin name with species author: ‘Pinus tabulaeformis Carr.’; in all further places it can be ‘Pinus tabulaeformis’ or ‘P. tabulaeformis’. If non-Latin name (Chinese pine) is used throughout the manuscript too (as in L82), it would be wise to put it first time together with Latin name

Author's Reply:Thanks very much! We have revised.

Reviewer’s Comment II: First mention counts separately in title, abstract, and text. Also, in keywords full species name is also required for search purposes. So please use full name ‘Pinus tabulaeformis Carr.’ in L20, 27, 83.

Reviewer’s Comment I: L17. Please rephrase as ‘PDSI is … factor limiting tree growth’

Author's Reply:Thanks very much! The emphasis here is on the limiting factor.

Reviewer’s Comment II: It is not about emphasis but just grammar correction.

Reviewer’s Comment I: L28. Keyword ‘transition zone’ is not appropriate. It is transition between what and what? Meaning of this term may vary greatly with context

Author's Reply:Thanks very much! ‘transition zone’ is a term in geography, representing the sensitivity of climate and other changes.

Reviewer’s Comment II: You should still clarify. There is also transition zone within the tree ring (between earlywood and latewood), so perhaps ‘climatic transition zone’ would be more concise term here?

Reviewer’s Comment I: L36. “These changes result in...’ – delete ‘be’ here

Author's Reply:It is ok.

Reviewer’s Comment II: It is grammar error, because passive voice ‘changes may be result’ reverses causal connection.

Reviewer’s Comment I: L46 rephrase as ‘precise-continuous dating of series’

Author's Reply:It is ok.

Reviewer’s Comment II: I don’t understand that is the meaning of this phrase ‘precise-continuous series for dating’, so please rephrase to make it more comprehensive.

Reviewer’s Comment I: L53 etc. Use ‘Mt.’ for single mountain peak (Mt. Shiren, I presume), and full word ‘Mountains’ for mountain ridges, ranges and larger systems, like ‘Funiu Mountains’. Now usage of abbreviation can confuse readers

Author's Reply:This is a good suggestion and will be used carefully in the future

Reviewer’s Comment II: Again, ignored in the manuscript.

Reviewer’s Comment I: L59-62. Sentence is difficult to understand

Author's Reply:It is ok, but this sentence is a bit long.

Reviewer’s Comment II: At least make it correct in grammatical sense then. I can understand long sentences fine, but usage of tenses is wrong here.

Reviewer’s Comment I: L64. What do you mean here: ‘north-south climatic factors’? The same in L89 with ‘east-west climate factors’

Author's Reply:Thanks very much! This expression means that research is carried out along a gradient.

Reviewer’s Comment II: Please tell it in the text of the manuscript too. Many readers are not familiar with such an expression.

Reviewer’s Comment I: L65-66. ‘climate-limiting factor’ – did you mean limiting tree growth climatic factor?

Author's Reply:It is ok.

Reviewer’s Comment II: ??? Phrase ‘climate-limiting factor’ literally means ‘factor that limits climate’! Please rephrase your sentence.

Reviewer’s Comment I: L78. ‘forests composed of temperate deciduous broad-leaved and coniferous’ – add word ‘trees’, or ‘species’, or both (‘tree species’)

Author's Reply:It is ok

Reviewer’s Comment II: Adjectives ‘broad-leaved and coniferous’ cannot be used without noun. Correct as suggested or rephrase otherwise.

Reviewer’s Comment I: L143-145. Phrases ‘southern meteorological station of Mt. Shiren’ and ‘northern meteorological station of Mt. Shiren’ are difficult to understand. It is written as if both stations are actually located on this mountain, just on the opposite slopes of it

Author's Reply:Thanks very much! We have revised.

Reviewer’s Comment II: This is somewhat better, but it is still written as if both stations are on the same mountain peak. Please, use ‘Funiu Mountains’. Also, see my comment above about ‘Mt.’ versus ‘Mountains’

Reviewer’s Comment I: L154. It should be ‘seasonal climatic response’, not ‘seasonal climate’ that is more stable than monthly one

Author's Reply:It is ok.

Reviewer’s Comment II: You did not describe above this moment any results on stability of seasonal climatic series. So please rephrase to avoid misleading readers.

Reviewer’s Comment I: Figure 4. Switch panels (a) and (b)

Author's Reply:Thanks very much! The labels are in the picture

Reviewer’s Comment II: Yes, but panel (a) should be on top. Numbering of panels within figure goes from top to bottom and usually from left to right.

Reviewer’s Comment I: Figure 5, L191 and further. ‘2.3a’ etc. – is it length of cycles in years? Not all readers are familiar with such abbreviation

Author's Reply:It is ok.

Reviewer’s Comment II: At least decipher this abbreviation on the first usage.

Reviewer’s Comment I: Figure 6. Please state window and step of moving correlations

Author's Reply:Thanks very much! We have added “based on window from January to December and 24 baselength”

Reviewer’s Comment II: Don’t understand what ‘24 baselength’ means. Please elaborate, possible describe procedure in Methods in more details.

Reviewer’s Comment I: L248. Use ‘extremely dry’ or ‘driest’, not mix of both

Author's Reply:It is ok. Here is the emphasis on the driest year in most extreme drought years.

Reviewer’s Comment II: It was request to correct grammar mistake, not stylistic choice.

Reviewer’s Comment I: Figure 8. The white rectangle mentioned in caption is not presented on figure itself

Author's Reply:Thanks very much! We have deleted the sentence.

Reviewer’s Comment II: But you should mark sampling site on this figure, with rectangle, dot or any other marker.

Author Response

Author's Reply see the attachment,please

Reviewer 4 Report

The revised version of the manuscript includes some improvements; however, most of the previously raised comments have not been addressed. Although the authors have responded to each comment, some of the responses seem that the comments were probably not well-comprehended. I provide below an elaboration of the comments, to make them even more clear:

  1. [Section 1] The paper is addressed to an international readership of the journal. Therefore the introduction should be more informative and the literature review should include overview for the tree species under study, the main aspects regarding the selected approach justifying its suitability for the specific case, as well as similar cases for pinus species related to the response to hydroclimatic parameters (indicatively see   https://doi.org/10.3390/f10090752, https://doi.org/10.3390/atmos11060554, https://doi.org/10.3390/f9070440 and/or other similar studies). Also, the use of the term “climate change” (e.g. L.53, 56, 57, 64) is not appropriate and should be restated (e.g. “climate” could be used in some of the cases), considering that dendrochronological studies can identify past climatic characteristics of region, which does not necessarily indicate climate “changes”.
  2. [Section 2] Although the original data are based on previous studies, some principal information (e.g. sampling time, sampling locations’ altitude, number of samples, timeseries, etc.) must be available in the text, in order the reader being able to evaluate the aspects of the further analysis. Additionally, the information included in the authors’ previous response should be also included in the manuscript. The above are rather important for explaining / discussing the identified relationships between tree growth and climatic factors (e.g. Tmax, P).
  3. [Fig. 2] The caption of the figure should be changed to indicate accurately the contents of the graphs, e.g. “Average monthly values of Tmax, T, Tmin, P and RH from Songxian (a) and Neixiang (b) meteorological stations during 1963-2016”.
  4. [Section 4] Apart from any description in Section 2 (which must be also elaborated), the discussion on the possible uncertainties of the analysis that must be added in this section is critical to understand the level of accuracy and reliability of the analysis. These uncertainties can be related to sampling issues (e.g. number of samples?), the actual association of the available stations / grid points to the sampling locations (distance, direction, topography, altitude differences, etc.); a commend on the uncertainties would be also appropriate in the conclusion. It must be also clarified and taken into account in the discussion and comparison to other studies, that the reconstructed drought conditions based on scPDSI refers specifically to May-June period, which does not necessarily represent the annual conditions or other seasons of each year. Regarding the spatial analysis of the reconstructed scPDSI, the previous response of the authors should be also added in the methodological section of the manuscript.

Author Response

(The authors gave the same response as above.)

Round 3

Reviewer 4 Report

The revised manuscript has been enhanced according to previous comments, providing clarifications on aspects of the methodological approach and analysis. However, there are still issues that have not been addressed and/or not sufficiently responded. More specifically:

  1. In the introductory section, the literature review on several aspects regarding the topic under study, as previously commented, is still quite poor and needs to be elaborated. Additionally, the issue regarding the use of the term “climate change” as it used in the introductory section (see previous comments), still remains.
  2. In section 2, the sampling site location has been added. However, Fig. 1 presents two locations as “tree-ring site” which is confusing to the reader. It is suggested to remove the second site that has not been used in the study from the map, or specifically explain this fact in the figure legend and the text. Furthermore, further details on sampling should be added (see also next comment).
  3. A previously commented, there must be some discussion on the uncertainties of the analysis (section 4). This does not mean that the analysis is not reliable, though its confidence level may vary. The previous reply by the authors on the matter (“tree rings are randomly sampled within a certain range, so I think that it statistically enhances the reliability of the conclusion”) does not clarify the above: apart from the fact that details on the sampling approach should been included in the text (see previous comments), the reliability (e.g. “we developed a 216years ring-width reliable chronology” [L.615]) of the analysis should be also properly justified. In any case, any response on the above should be added in the text.

Author Response

Author's Reply to the Reviewer  see  attachment.
